# Interleukin-35 Prevents the Elevation of the M1/M2 Ratio of Macrophages in Experimental Type 1 Diabetes

**DOI:** 10.3390/ijms23147970

**Published:** 2022-07-19

**Authors:** Zhengkang Luo, Charlotte Soläng, Rasmus Larsson, Kailash Singh

**Affiliations:** Department of Medical Cell Biology, Uppsala University, 75123 Uppsala, Sweden; zhengkang.luo@mcb.uu.se (Z.L.); charlotte.solang@gmail.com (C.S.); rasmus.larsson.7292@student.uu.se (R.L.)

**Keywords:** type 1 diabetes, macrophages (M1 and M2), pDCs, IL-35

## Abstract

Macrophages play an important role in the early development of type 1 diabetes (T1D). Based on the phenotype, macrophages can be classified into pro-inflammatory (M1) and anti-inflammatory (M2) macrophages. Despite intensive research in the field of macrophages and T1D, the kinetic response of M1/M2 ratio has not been studied in T1D. Thus, herein, we studied the M1 and M2 macrophages in the early development of T1D using the multiple low dose streptozotocin (MLDSTZ) mouse model. We determined the proportions of M1 and M2 macrophages in thymic glands, pancreatic lymph nodes and spleens on days 3, 7 and 10 after the first injection of STZ. In addition, we investigated the effect of IL-35 in vivo on the M1/M2 ratio and IL-35^+^ plasmacytoid dendritic cells in diabetic mice and in vitro on the sorted macrophages. Our results revealed that the M1/M2 ratio is higher in STZ-treated mice but this was lowered upon the treatment with IL-35. Furthermore, IL-35 treated mice had lower blood glucose levels and a higher proportion of IL-35^+^ cells among pDCs. Macrophages treated with IL-35 in vitro also had a higher proportion of M2 macrophages. Together, our data indicate that, under diabetic conditions, pro-inflammatory macrophages increased, but IL-35 treatment decreased the pro-inflammatory macrophages and increased anti-inflammatory macrophages, further suggesting that IL-35 prevents hyperglycemia by maintaining the anti-inflammatory phenotype of macrophages and other immune cells. Thus, IL-35 should be further investigated for the treatment of T1D and other autoimmune disorders.

## 1. Introduction

Type 1 diabetes (T1D) is an autoimmune disease featured by pancreatic β-cell destruction by immune attacks. The absolute deficiency of insulin production leads to hyperglycemia. CD4 and CD8 autoreactive T-cells, macrophages and/or dendritic cells (DCs) have been found to be the first infiltrating immune cells in islets at the early stage of T1D development as reviewed previously [1]. In line with this, we also reported earlier that DCs increased at the early development of T1D [2]. The homeostasis of the immune system is mediated by several types of regulatory immune cells, such as regulatory B (Breg) cells and regulatory T (Treg) cells. Breg and Treg prevent immune cells from attacking self-tissues through a variety of mechanisms, including secreting suppressive cytokines such as Interleukin-35 (IL-35). We have previously reported that the plasma levels of IL-35 were reduced in patients with T1D compared with healthy individuals, and IL-35 treatment prevented the development of hyperglycemia and reversed established hyperglycemia in mouse models of T1D [3,4]. In addition, T1D patients with remaining c-peptides had a higher circulating level of IL-35, indicating that IL-35 may prevent β-cell destruction [5]. We have also reported that circulating IL-35 levels are lowered in patients with type 2 diabetes (T2D) compared with healthy controls [3].

Macrophages display plastic characteristics and their involvement has been reported in many autoimmune diseases including T1D [6]. Conventionally, macrophage phenotypes can be classified as M1 (classically activated; pro-inflammatory) and M2 (alternatively activated; anti-inflammatory) [7]. M1 macrophages act as pathogen killers and can produce pro-inflammatory cytokines such as IL-1β, IL-6, IL-12, IL-23 and tumor necrosis factor alpha (TNF-α) [8]. On the contrary, M2 macrophages are considered as anti-inflammatory and can produce cytokines such as IL-10 and TGF-β and the enzyme Arginase 1 (Arg1) [7,8,9]. Although this simplified model does not fully represent the complexity of macrophage plasticity, an imbalance of M1/M2 was found in several autoimmune diseases [10], and increase in the M1 phenotype has been related to the development of T1D [11]. IL-1β produced by M1 triggers β-cell destruction, and earlier, our group reported a complete protection of β-cells using IL-1 cytokine trap [12], indicating that cytokines produced by M1 play a crucial role in pathogenesis of T1D. Plasmacytoid dendritic cells (pDCs) also contribute in the early development of T1D by producing IFN-α [13,14], which is also supported by our earlier finding that pDCs increased in the development of experimental T1D [2].

We have reported that IL-35 treatment prevents and reverses hyperglycemia in T1D mouse models by enhancing responses of Breg and Treg cells [4,15]. In addition, Koda et al. have shown that pDCs protect against acute liver injury via IL-35 [16]. Therefore, we intended to investigate the effects of IL-35 treatment on macrophage polarization and response of pDCs in an experimental T1D, mouse model: multiple low dose streptozotocin (MLDSTZ). Herein, we used the MLDSTZ mouse model of T1D since in this model mice develop hyperglycemia and insulitis gradually [4,17]. Thus, this model provides us with the opportunity to study the kinetic responses of immune cells in early development of T1D.

## 2. Results

To investigate the kinetic response of macrophages in the early development of experimental autoimmune diabetes. Male CD-1 mice were treated with MLDSTZ intraperitoneally for five consecutive days. Thymi, pancreatic draining lymph nodes (PDLNs) and spleens were collected on days 3, 7 and 10 after the first STZ injection, respectively. On these days (days 3, 7 and 10), STZ-treated mice did not develop hyperglycemia, although mice had higher glucose levels compared to vehicle-treated mice on day 10 [2]. Single cells were prepared for flow cytometry. The thymus is one of the central organs of the immune system, and Liu et al. reported the presence of different subsets of macrophages in thymic glands [18]. We therefore determined the proportions of M1 and M2 in thymic glands. PDLNs were investigated to determine the local immune response and splenocytes were isolated to study the systemic immune response. No difference in the proportions of macrophages were observed in all three organs: thymi, PDLNs and spleen (Figure 1A). To characterize the phenotype of macrophages, we next used the pro-inflammatory cytokine TNF-α and Arginase 1 (Arg1) to determine M1 and M2 macrophages. No difference in the proportions of TNF-α^+^ macrophages were found on day 3 and day 7, but we found an increase in the proportions of TNF-α^+^ macrophages on day 10 in the spleens of STZ mice (Figure 1B). On the other hand, the proportions of Arg1^+^ macrophages decreased in the spleens of STZ mice on day 3, but these proportions were increased in the thymi and spleens in STZ mice on day 10 (Figure 1C).

To further demonstrate macrophage polarization, the M1/M2 ratio was calculated using TNF-α^+^ and Arg1^+^ cell proportions among macrophages. The M1/M2 ratio was elevated in the spleens of STZ mice on day 3 and 7 compared with vehicle mice (Figure 2), indicating a skewing of macrophages toward M1 in the early development of experimental T1D. No statistical difference was observed in the thymi and PDLNs (Figure 2).

The proportions of Arg1^+^TNF-α^+^ macrophages in the spleens of STZ mice on day 3 and in all organs of STZ mice on day 7 and day 10 increased (Figure 3A). We also found an increase in the proportions of TNF-α^+^ cells among Arg1^+^ macrophages in the thymi of STZ mice on day 7, and in the PDLNs and spleens of STZ mice on days 3, 7 and 10 (Figure 3B). Taken together, these results demonstrate an elevated TNF-α production in M2 macrophages in STZ mice, further supporting the polarization toward M1 in the early development of experimental T1D.

Next, we determined the proportions of M1 and M2 macrophages in diabetic mice (blood glucose levels are shown in Appendix A). Moreover, the immune cells were restimulated with PMA, ionomycin and Brefeldin A with and without LPS, and a group of immune cells was not at all restimulated. We had these three groups, as macrophages are very plastic cells and tend to switch their phenotype upon stimulation. At the resting stage (no restimulation), the proportion of M1 macrophages was not altered in diabetic mice (Figure 4). Similar results were observed when the immune cells were restimulated with or without LPS (Figure 4).

To investigate the effect of IL-35 treatment on macrophages in vivo, male CD-1 mice were treated with PBS or recombinant IL-35 after MLDSTZ injections. Mice that were treated with PBS induced hyperglycemia, but IL-35 treatment reduced the blood glucose elevation after STZ injection (Figure 5A). Next, we found that the proportions of macrophages were lower in the spleens of IL-35-treated mice, but no difference was found in PDLNs (Figure 5B). The proportions of TNF-α^+^ cells among macrophages were lower in the PDLNs of IL-35 mice (Figure 5C). On the contrary, the proportions of Arg1^+^ cells among macrophages were higher in the spleen and PDLN of IL-35-treated mice (Figure 5D). Likewise, the M1/M2 ratio was determined to demonstrate macrophage polarization. We found that it was the M1/M2 ratio was higher in both spleens and PDLNs of STZ mice (Figure 5E), suggesting that macrophages were more skewed toward M2 phenotype after IL-35 treatment.

Next, we investigated the expression levels of Ebi3 and IL-12p35 (subunits of IL-35) in macrophages. Interestingly, we found that the mean fluorescence intensities (MFIs) of both Ebi3 and IL-12p35 were lowered in the spleens of STZ-treated mice on day 3 (Figure 6A,B). These results indicate that IL-35^+^ macrophages are impaired in spleens of STZ mice in the early stage of T1D development. Subsequently, we studied whether IL-35 treatment affected the expression of IL-35 in macrophages. However, we did not find any alteration of IL-35 expression upon the systemic treatment with IL-35 (Figure 6C).

Next, we sorted CD11b^+^F4/80^+^ macrophages to investigate the direct effect of IL-35 on macrophages. CD11b^+^F4/80^+^ macrophages were treated with or without recombinant IL-35 (10ng/mL) overnight. IL-35-treated macrophages had lowered proportions of M1 (TNF-α^+^ or MHC-II^+^ cells among macrophages) and a higher proportion of M2 (Arg1^+^ cells among macrophages) macrophages (Figure 7A–C). Together, these data demonstrate that the effect of IL-35 is direct on macrophages in lowering the M1/M2 ratio. 

Similar to macrophages, pDCs are also increased in the early stage of T1D development [2]. Furthermore, it has been reported that pDCs can produce IL-35 and play an important role in suppressing inflammation in acute liver disease [16]. Thus, we studied the expression of IL-35 in pDCs in experimental T1D. Systemic IL-35 treatment did not significantly increase the expression of IL-35 subunits in pDCs (Figure 8A,B). However, the proportion of IL-35^+^ cells among pDCs was higher in IL-35-treated mice (Figure 8C).

## 3. Discussion

Our current study demonstrates that macrophages are skewed toward M1 phenotype in the early development of experimental T1D, and IL-35 treatment promotes M2 polarization. 

TNF-α has been related to the pathogenesis of T1D, as pancreas-resident macrophages have been found to produce TNF-α [11]. TNF-α derived from Th1 cells can either induce β cell damage directly or activate M1 macrophages to produce reactive oxygen species (ROS), TNF-α and IL-1β [19]. In addition, the blockade of TNF-α prevented the development of diabetes after the transfer of lymphocytes from NOD mice [20]. Earlier, our group has reported that a TNF-α inhibitor (MDL 201.449A) prevented the development of diabetes in MLDSTZ mice [21]. Therefore, reducing TNF-α production from immune cells is beneficial for the prevention of T1D development in MLDSTZ mouse model. Previous attempts to treat collagen-induced arthritis in mice with IL-35 resulted in the decrease in TNF-α at both gene and protein levels [22]. However, no study has been carried out to measure the effect of IL-35 on TNF-α production in the T1D scenario. Herein, we report that even though the proportions of TNF-α^+^ cells among macrophages in the spleen remained unaltered after IL-35 treatment, they were reduced in the PDLN, suggesting that the protective role of IL-35 via inhibiting TNF-α production works in a more local manner. With regard to the role of Arg1, studies show conflicting results. Ahn et al. postulated that Arg1, which was partly produced from M2 macrophages, modulated the neuroinflammation in experimental autoimmune encephalomyelitis (EAE) [23]. On the other hand, increased vascular Arg1 levels have been related to impaired vascular functions in T1D rodent models [24,25]. Furthermore, female NOD mice with Arg1 inhibition had less incidence of developing hyperglycemia than PBS-treated NOD mice [26]. However, these results do not disagree with our data as they focused on Arg1 with a vascular endothelial origin, and the NOD mice in the other study received systemic Arg1 inhibition. A study using macrophage-specific knockout of Arg1 revealed the protective role of M2 macrophages in chronic schistosomiasis [27], but no such study has been performed in T1D. 

Early studies on the inactivation or depletion of macrophages revealed the necessity of macrophages in the autoimmune attack to islets in NOD mice [28,29]. The islet resident macrophages of NOD mice were found to have a less immunoregulatory phenotype [30]. The bone-marrow-derived macrophage from diabetic mice presented a more M1 phenotype [31]. Moreover, the adoptive transfer of M2 macrophages prevented T1D in NOD mice [32]. These studies have demonstrated the significance of macrophage phenotype/polarization in T1D research. A study on EAE has shown that a shift of M1 to M2 was found in the alleviation of EAE severity [33]. In another study, in which wound healing in experimental T1D was improved after a knockout of a leukotriene, a shift of macrophages from M1 to M2 was observed [34]. Taken together, previous studies and our current data showing macrophages being skewed toward M1 in T1D animal models and a shift to M2 could be found in the prevention of experimental T1D could bring more insights on the polarization of macrophages and their roles in the development of T1D. 

We have previously shown that IL-35 treatment effectively prevented the development of experimental T1D in mouse models by modulating Treg and Breg responses [4,15]. Despite intensive studies on the roles of IL-35 in Breg and Treg cells, the effect of IL-35 on macrophages in the context of autoimmunity has not been well characterized. Zhang et al. reported that IL-35 treatment reduced local macrophage infiltration and the M1/M2 ratio in experimental psoriasis [35]. Our current finding that IL-35 treatment reduced the M1/M2 ratio in experimental T1D is in line with previous reports. Yet, we do not know whether the shift in M1/M2 ratio is due to the change to Breg and Treg cells or a consequence of the direct response of macrophages to IL-35, although our in vitro experiments revealed that it is due to a direct effect of IL-35 on macrophages. In line with this, Koda et al. found that IL-35 producing Treg cells suppress ongoing immune assault in acute liver injury through pDCs [16]. Herein, we also found a higher proportion of IL-35^+^ cells among pDcs in IL-35 treated mice. 

## 4. Materials and Methods

### 4.1. Animals

All experiments including laboratory animals were approved by the regional ethical committee of the Uppsala County. Male CD-1 mice were obtained from Charles River (Hannover, Germany). The mice were thereafter housed in the animal facility at the Biomedical Center, Uppsala, Sweden.

Male CD-1 mice were injected with STZ (Sigma-Aldrich, St Louise, MO, USA; 40 mg/kg body weight) dissolved in 200 µL saline solution for five consecutive days to induce autoimmune hyperglycemia [17]. Mice in the treatment groups further received injection with 200 µL phosphate-buffered saline (PBS) or PBS containing IL-35 (mouse recombinant IL-35, Chimerigen, Liestal, Switzerland; 0.75 μg/day) for eight days. Blood was obtained from the tail of the mice, and the blood glucose levels were measured using a blood glucose meter (FreeStyle Freedom Lite, Abbott, Chicago, IL, USA).

### 4.2. Single Cell Preparation

Single cells from thymic glands, pancreatic draining lymph nodes (PDLNs) and spleens were prepared as previously described [36,37]. In short, thymi and spleens were collected and squeezed with a pair of tweezers to release cells. Cells were then lysed with 0.2 M NH_4_Cl and resuspended in Hank’s balanced salt solution (HBSS; Statens veterinärmedicinska anstalt, Uppsala, Sweden). PDLNs were grinded with a pair of tweezers on a sterile metal mesh and were then washed and resuspended with RPMI-1640 (Sigma-Aldrich).

### 4.3. Monoclonal Antibody Staining and Flow Cytometry

Single cells from thymic glands, PDLNs and spleen were stained with the following surface antibodies: CD11b (M1/70, BioLegend, San Diego, CA, USA), F4/80 (BM8, BioLegend), MHC-II (M5/114.15.2, BioLegend), CD11c (N418, BioLegend), B220 (RA3-6B2, BioLegend) and PDCA-1 (927, BioLegend). The cells were then permeabilized and fixed overnight at 4 °C with Fixation and Permeabilization Buffer (eBioscience, San Diego, CA, USA). The next morning, single cells were stained with antibodies to Arginase 1 (A1exF5, ThermoFisher, Auburn, AL, USA), TNF-α (MP6-XT22, BioLegend), Ebi3 (355022, R&D Systems, Minneapolis, MN, USA) and IL-12p35 (27537, R&D Systems). Fc block (#553142 BD BioSciences, San Jose, CA, USA) was used for both surface and intracellular staining. Fixable Viability Dye eFluor^TM^ 780 (Thermofisher) was used for detecting live cells. All the samples were analyzed using a BD LSR Fortessa at the BioVis Platform (Uppsala University, Uppsala, Sweden). Data from flow cytometry were analyzed using the Flowlogic version 8.6 software (Inivai Technologies, Mentone, Australia) and FlowJo (Ashland, OR, USA). Gating strategies used for analysis are shown in Appendix A.

### 4.4. Cell Sorting

Splenocytes from male CD-1 mice were stained with CD11b and F4/80 antibodies, and CD11b^+^F4/80^+^ macrophages were sorted using BD FACSMelody at the BioVis Platform, Uppsala University. 

### 4.5. Cell Culture

Splenocytes from MLDSTZ-induced diabetic male CD-1 mice were cultured at 37 °C for 5 h with/without LPS (Sigma-Aldrich, 10 ng/mL) and with/without cell activation cocktail (#423304, BioLegend, containing PMA, ionomycin and Brefeldin A, 2 µL per mL cell suspension). 

Sorted macrophages were cultured at 37 °C for 24 h with LPS (10 ng/mL) and with or without IL-35 (recombinant IL-35, PeproTech, Cranbury, NJ, USA, 10 ng/mL). Cell activation cocktail (2 µL per mL cell suspension) was added 5 h prior to cell collection. The cells were then stained for Arg-1, CD11b, F4/80, MHC-II, and Fixable Viability Dye eFluor^TM^ 780 (Thermofisher) and acquired on BD LSR Fortessa flow cytometry. 

### 4.6. Statistical Analysis

GraphPad Prism version 7.02 and 9.4.0 (GraphPad software San Diego, CA, USA) were used for all the statistical analysis. Repeated two-way ANOVA followed by Sidak’s test and unpaired *t*-test were performed to compare differences between groups. *, ** and *** denote *p* < 0.05, *p* < 0.01 and *p* < 0.001, respectively.

## Figures and Tables

**Figure 1 ijms-23-07970-f001:**
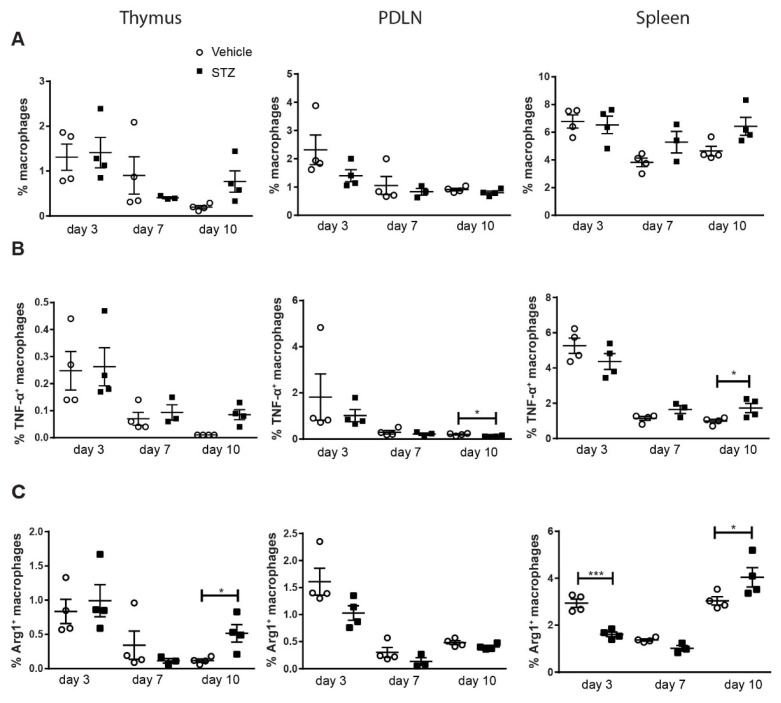
The kinetics of macrophage polarization. Male CD-1 mice were injected intraperitoneally with saline or low dose STZ (40 mg/kg body weight) for 5 consecutive days. Single cells were prepared from thymi, PDLNs and spleens. (**A**) The proportions of macrophages. (**B**) The proportions of TNF-α^+^ macrophages. (**C**) The proportions of Arg1^+^ macrophages. Results are expressed as means ± SEM. Unpaired *t*-tests were performed for comparisons. * and *** denote *p* < 0.05 and *p* < 0.001.

**Figure 2 ijms-23-07970-f002:**
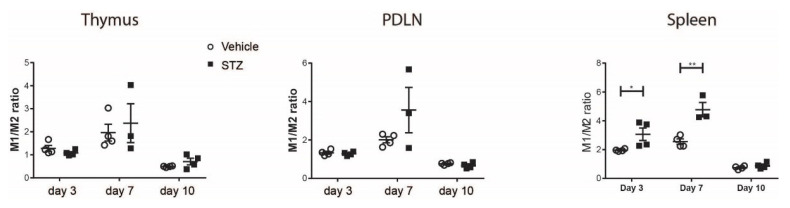
The kinetics of M1/M2 ratio. M1/M2 ratio in the thymi, PDLNs and spleens of vehicle-treated or STZ-treated mice. Results are expressed as means ± SEM. Unpaired *t*-tests were performed for comparisons. * and ** denote *p* < 0.05 and *p* < 0.01.

**Figure 3 ijms-23-07970-f003:**
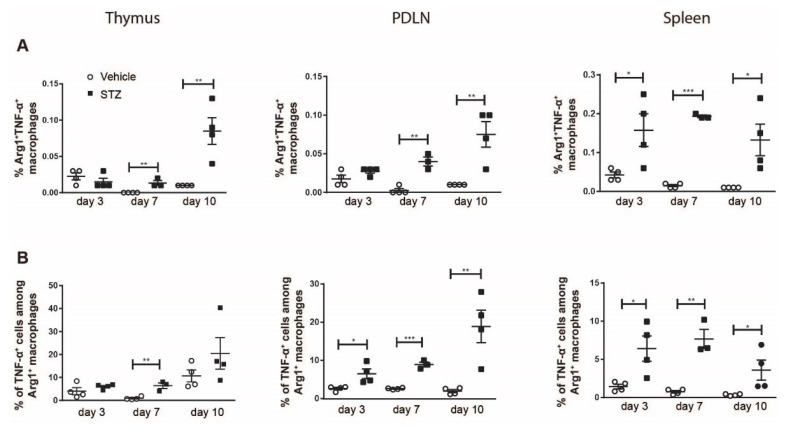
The kinetics of TNF-α production in M2 macrophages. Single cells were prepared from the thymi, PDLNs and spleens of vehicle or STZ treated mice. (**A**) The proportions of TNF-α^+^ M2 macrophages. (**B**) The proportions of TNF-α^+^ cells among M2 macrophages. Results are expressed as means ± SEM. Unpaired t tests were performed for comparisons. *, ** and *** denote *p* < 0.05, *p* < 0.01 and *p* < 0.001.

**Figure 4 ijms-23-07970-f004:**
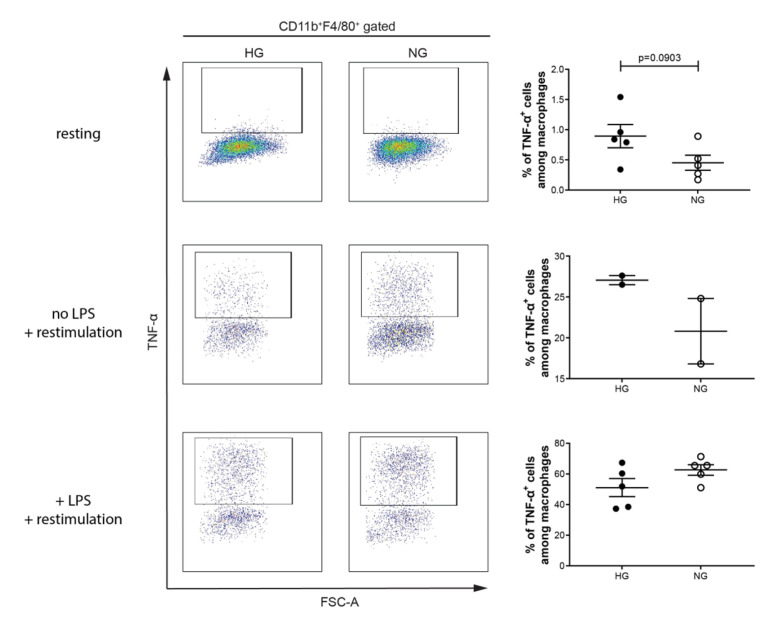
Percentage of M1 macrophage is not altered in diabetic mice. Splenocytes of MLDSTZ induced diabetic mice (HG; high glucose levels) or non-diabetic mice (NG; normal glucose levels) were stained without restimulation (resting), restimulated with or without LPS for 4–5 h, and then stained with antibodies and then the cells were analyzed by flow cytometry. Live cells were gated for CD11b^+^F4/80^+^ macrophages and then these cells were further gated for determining the percentages of TNF- α^+^ cells among CD11b^+^F4/80^+^ macrophages using FlowJo software. Results are expressed as means ± SEM. Unpaired *t*-tests were performed for comparisons.

**Figure 5 ijms-23-07970-f005:**
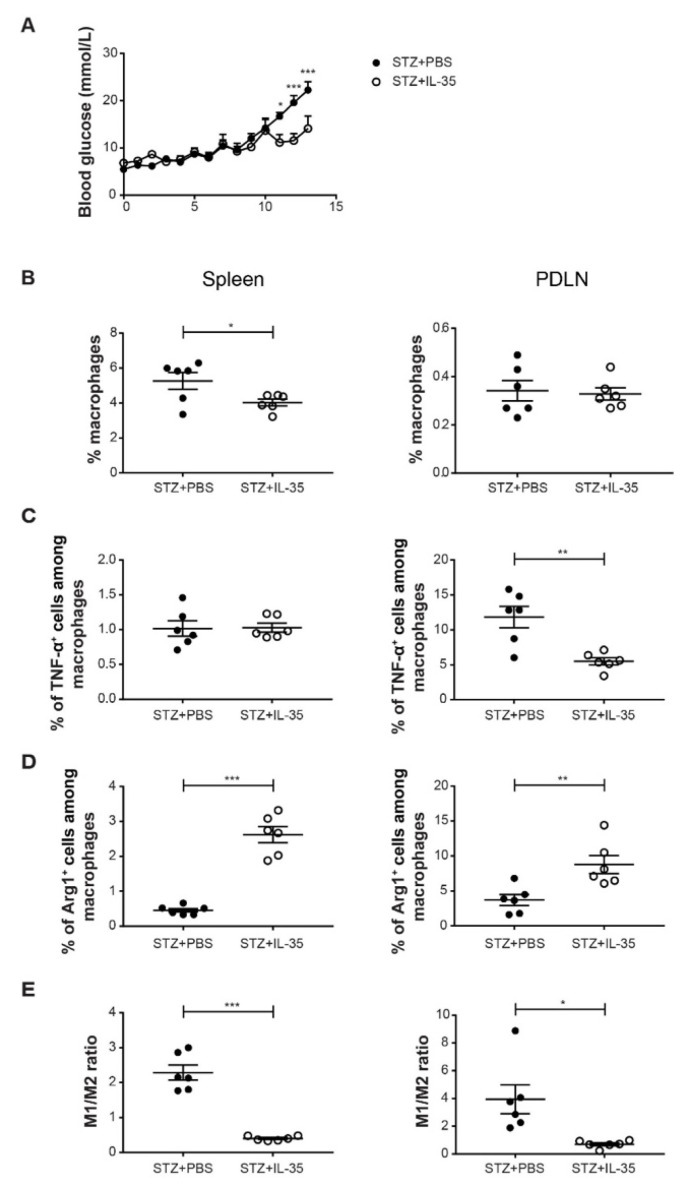
IL-35 treatment reduced the M1/M2 ratio in STZ mice. Male CD-1 mice were injected intraperitoneally with low doses of STZ (40 mg/kg body weight) for 5 consecutive days. For the next 8 days, mice were injected daily with PBS or recombinant mouse IL-35 (0.75 μg/day). Blood glucose levels of the mice were measured every day from day 5 after the first STZ injection (**A**). The mice were killed on day 13. Single cells were prepared from harvested PDLNs and spleens. (**B**) The proportions of macrophages. (**C**) The proportions of TNF-α^+^ cells among macrophages. (**D**) The proportions of Arg1^+^ cells among macrophages. (**E**) M1/M2 ratio. Results are expressed as means ± SEM. Repeated two-way ANOVA followed by Sidak’s test and unpaired *t*-tests were performed for comparisons. *, ** and *** denote *p* < 0.05, *p* < 0.01 and *p* < 0.001.

**Figure 6 ijms-23-07970-f006:**
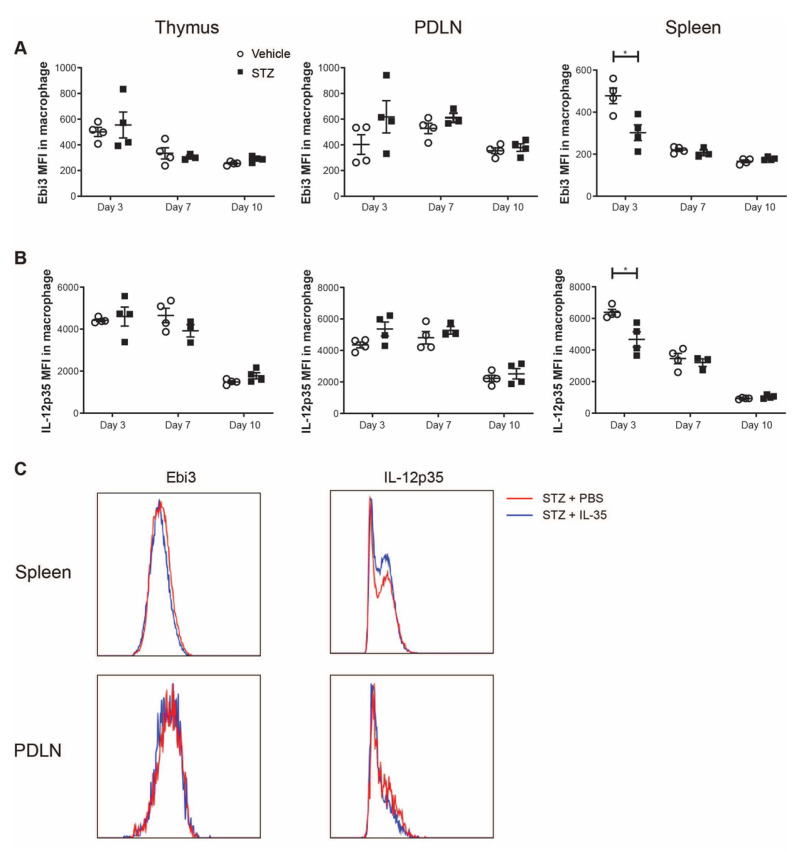
Mean fluorescence intensity (MFI) of Ebi3 and IL-12p35 in macrophages. Male CD-1 mice were injected intraperitoneally with low doses of STZ (40 mg/kg body weight) or saline for 5 consecutive days and were killed on days 3, 7 and 10 after the first injection. (**A**,**B**) The MFI of Ebi3 and IL-12p35 in macrophages in the thymus, PDLN and spleen. STZ mice received IL-35 (0.75 μg per day) and PBS injections for 8 days and were killed on day 14. (**C**) The representative histogram of fluorescence intensity of Ebi3 and IL-12p35 in macrophages in the spleen and PDLN. Results are expressed as means ± SEM. Unpaired *t*-tests were performed for comparisons. * denotes *p* < 0.05.

**Figure 7 ijms-23-07970-f007:**
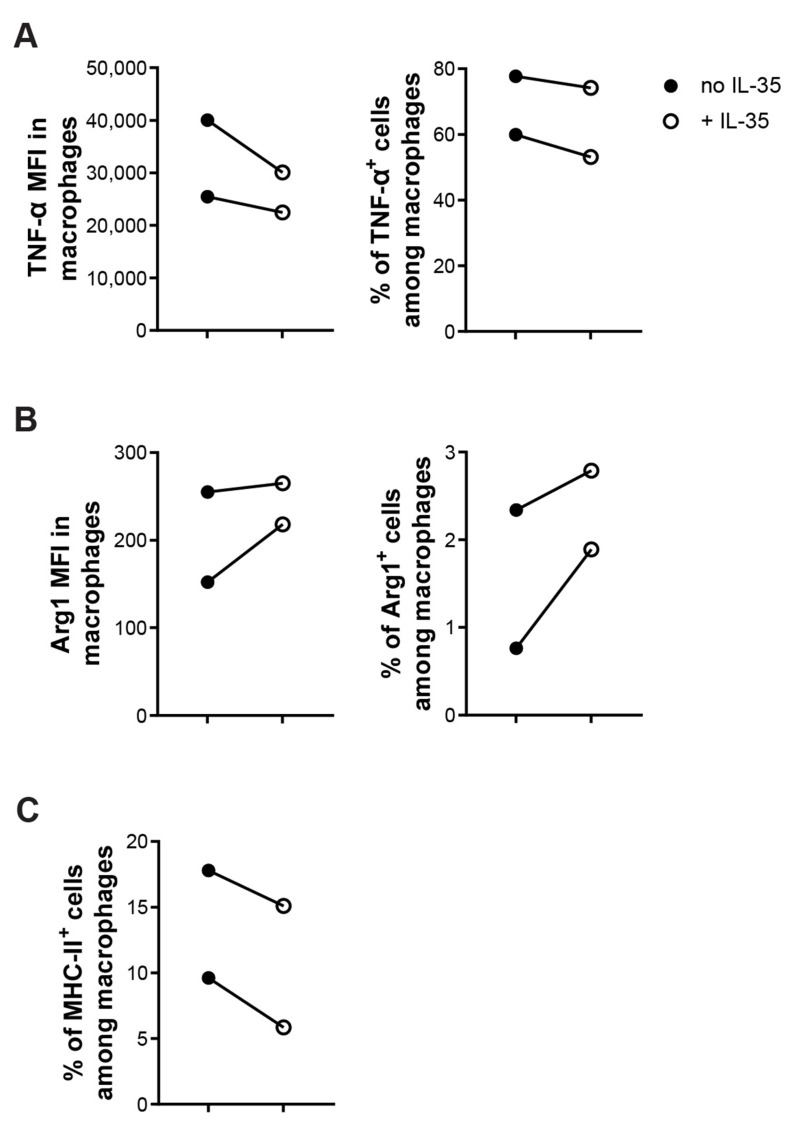
IL-35 treatment effects directly on macrophages to lowered M1/M2 ration in vitro. Sorted CD11b^+^F4/80^+^ macrophages were created with LPS with or without recombinant IL-35 (10 ng/mL) for 24 h in a 96-well plate. Harvested cells were stained and cells were analyzed by flow cytometry. (**A**) MFI of TNF-α among macrophages and percentage of TNF-α^+^ cells among macrophages. (**B**) MFI of Arg1 among macrophages and percentage of Arg1^+^ cells among macrophages. (**C**) Percentage of MHC-II^+^ cells among macrophages. Each experiment was repeated twice. Paired *t*-tests were performed for comparisons.

**Figure 8 ijms-23-07970-f008:**
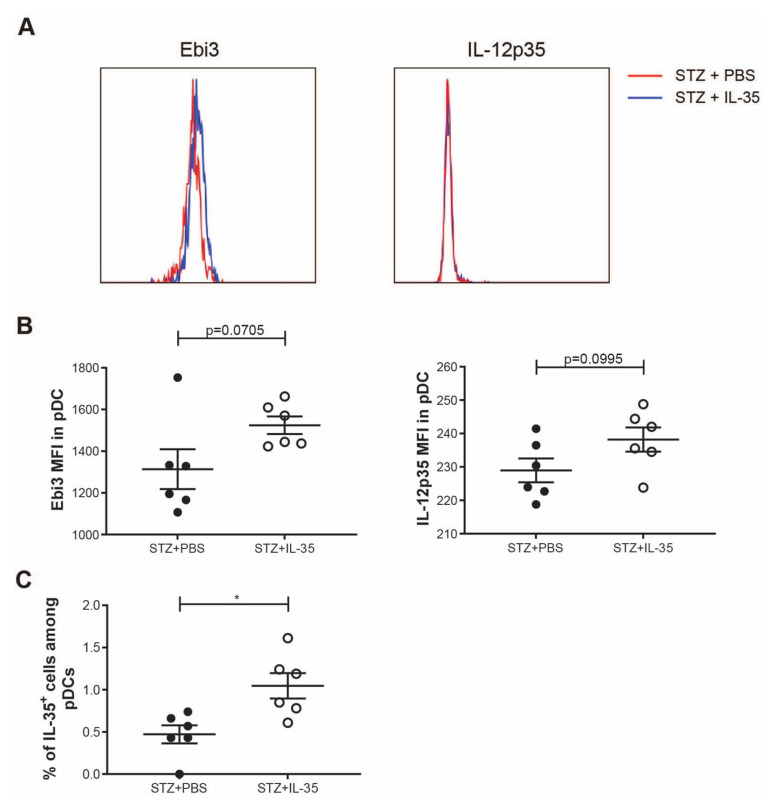
IL-35 expression in pDCs. STZ mice received IL-35 (0.75 μg per day) or PBS treatment for the following 8 days and were killed on day 14. Single cells from the spleen were prepared. (**A**) The representative histogram of fluorescence of Ebi3 and IL-12p35 in pDCs. (**B**) The MFI of Ebi3 and IL-12p35 in pDCs. (**C**) The proportion of IL-35^+^ cells among pDCs. Results are expressed as means ± SEM. Unpaired *t*-tests were performed for comparisons. * denotes *p* < 0.05.

## Data Availability

Details of all experiments, including data and material used for performing this study, will be made accessible. Data are either included in the manuscript or available upon request.

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
