# Peer review of "Interleukin-35 Prevents the Elevation of the M1/M2 Ratio of Macrophages in Experimental Type 1 Diabetes"

_ijms, 2022, doi:10.3390/ijms23147970_

Round 1
Reviewer 1 Report
In the paper, the authors have studied the presence of M1 and M2 macrophages in T1D induced by multiple low dose streptozotocin (MLDSTZ) in mice. In addition, administration of IL-35 to T1D mice lowered blood glucose levels and M1/M2 ratio. Also, macrophages treated with IL-35 in vitro showed an M2 polarization. Overall, their data indicate that under diabetic conditions, the pro-inflammatory macrophages are increased, and that IL-35 treatment is able to decrease the percentage of the pro-inflammatory macrophages while increasing the percentage of anti-inflammatory macrophages.
This is a resubmission of a previously rejected paper. The issues raised in the previous versions have been acknowledged.
Author Response
We like to thank the reviewer for his/her time and providing very good feedback on our manuscript.
Reviewer 2 Report
Minor grammatical points:
Line:
2 prevents elevation of the M1/M2 ratio
23 , further
34 to be the
35 this,
38 cells and
47 display plastic
55 , an imbalance
56 and increase of the M1
57 , and earlier
66 al. have shown
69 T1D,
78-79 had higher
79 vehicle-treated
80 The thymus
81 . We
86 the pro-inflammatory
103 skewing of
122 Next,
125 groups,
126 , the proportion
126 macrophages was not
128 without LPS
133 with antibodies
134 were analysed by flow
138 were treated
138 IL-35 after
138 Mice which were
145 that it was
151 reduced the M1/M2
152 doses of STZ
164 the early
166 treatment with IL-35
170 with low doses of
184 were ctreated with LPS with or
185 were analysed by flow
197 IL-35-treated
214 that a TNF
216 in the MLDSTZ
217 attempts
230 and the NOD
231 macrophage-specific
255 in line with this,
256 al. have found that
257 . Herein
258-259 delete sentence "Thus, .....versa."
291 were stained
291 delete "intracellular"
291 antibodies to
306 MLDSTZ-induced
343 flow cytometry.
Author Response
We like to thank the reviewer for his/her time and providing very good feedback on our manuscript.This manuscript is a resubmission of an earlier submission. The following is a list of the peer review reports and author responses from that submission.
Round 1
Reviewer 1 Report
Please see accompanying pdf.
The streptozotocin model of diabetes type 1 should be descibed in more detail, especially in relation to current models of diabetes type 1 development.

Reviewer 2 Report
In the paper, the authors have studied the presence of M1 and M2 macrophages in T1D induced by multiple low dose streptozotocin (MLDSTZ) in mice. In addition, administration of IL-35 to T1D mice lowered blood glucose levels and M1/M2 ratio.
Issues to address
The methods used for the characterization of M1 and M2 macrophages need to be improved. There are several markers that better define these populations, for instance CD80, CD86, CD64, CD206….
The use of IC staining of TNFalpha is not reliable, as cytokines are synthetized and readily secreted and hence it is difficult to detect them, without the restimulation of the cells.
When working with monocyte/macrophages, it is important to use Fc blocking agents, otherwise the data are not reliable.
We could not find the supplementary materials and hence the gating strategy could not be verified.
It would improve the quality of the manuscript the inclusion of the in vitro effect of IL35 in macrophage polarization. Indeed, the administration of IL35 in vivo may alter the course of the disease and the macrophage polarization by acting not only on the macrophage compartment but also on the adaptive immunity.
Please explain why thymic macrophages have been investigated in T1D.
In the data presented in Figure 4, the negative control group (i.e. healthy, not STZ challenged mice) should be included.
Round 2
Reviewer 2 Report
The authors have not addressed any of the issues raised in the first round of revision.
The M1 and M2 subpopulations should better be defined by a wider panel of markers.
The methodology used for the staining is not correct, as it is missing the use of Fc Block.
The ICS TNFalpha is not reliable, as it should be performed on restimulated cells along with GolgiStop treatment.
The direct effect of IL35 on macrophage polarization has not been provided. It could be easily done in vitro.
The negative control group is missing in figure 4.
Author Response
The authors have not addressed any of the issues raised in the first round of revision.
Authors We like to thank the reviewer for his/her time and providing very good feedback on our manuscript. We apologize for inconvenience; however, we did provide the point-by-point response letter.
The M1 and M2 subpopulations should better be defined by a wider panel of markers.
Authors: We are very regret that we did not put all of these antibodies in the panel to determine the M1 and M2 macrophages. Nevertheless, a majority of polished studies have used F4/80, CD11b, TNF-alpha, Arg-1, thus, we also used these markers only to defined M1 and M2 population.
The methodology used for the staining is not correct, as it is missing the use of Fc Block.
Authors: We are very regret that we did not Fc blocking agents, as our pilot study indicated that we don’t need such reagents as we didn’t find any differences in frequency of particular myeloid cells whether we stain with or without Fc block.
The ICS TNFalpha is not reliable, as it should be performed on restimulated cells along with GolgiStop treatment.
Authors: We did not stimulate the cells since immune cells use to have a very plastic in nature. In the present study our aim was to identify native macrophages response and after stimulation the cell response may not represent what is really going on in different mouse, thus we determined cell proportions directly after single cell preparation without stimulation. This has been clarified in the present version of this manuscript
Line numbers 204-207
” In the present study we didn’t stimulate cells for intracellular cytokine staining, as we wanted to study the cytokine response in native condition. Upon stimulation macrophage tands to change their phenotype.”
The direct effect of IL35 on macrophage polarization has not been provided. It could be easily done in vitro.
Authors: We agree that this a very good suggestion. But we think that this is beyond the scope of the present manuscript. Indeed, in vitro experiments will be most useful in the further exploration of the role of IL-35 in polarization of macrophage.
The negative control group is missing in figure 4.
Authors: This is a good point but, in this study, we aimed to determine the proportion of m1 and M2 in STZ mice treated with PBS and STZ mice treated with IL-35. In addition, in a previous study we have reported data from mice treated with Vehicle + PBS (that are the real negative ones). We did not find any differences of immune cell response between this group and STZ + IL35 treated group. (https://www.mdpi.com/1422-0067/22/23/12988)
Figure 4. IL-35 treatment prevents hyperglycemia in the MLDSTZ mouse model and reverses the decreased Breg cell proportions. Male CD-1 mice were injected with saline or low doses of STZ for 5 consecutive days. For the next 8 days, mice received saline further, were injected daily with PBS, and STZ mice were injected daily with PBS or recombinant mouse IL-35 (0.75 μg/day). Blood glucose levels of the mice were measured every day from day 5 after the first STZ injection, and the percentages of diabetes-free mice were measured. (A) The mice were killed on day 13. Single cells were prepared from removed thymic glands, PDLNs, and spleens. The proportions of (B) CD19+CD1d+CD5+ Breg cells, (C) Breg cells among CD19+ cells and CD19+cells, and (D) IL-35+ cells among CD19+CD1d+CD5+ Breg cells were determined by flow cytometry. Results are expressed as means ± SEM from two experiments (n = 2–3 mice/group/experiment). Repeated two-way ANOVA followed by Dunnett’s test, one-way ANOVA followed by Tukey’s test, and unpaired t-tests were performed for comparisons. *, **, and *** denote p < 0.05, p < 0.01, and p < 0.001, respectively.

Round 3
Reviewer 2 Report
The ICS TNFalpha is not reliable and it needs to be performed on cells restimulated and treated with GolgiStop. The restimulation is usually carried over for 3-4 hrs and in this short time-frame, no changes in the macrophage phenotype will occur.
The evaluation of the direct effect of IL35 on macrophage polarization is not trivial, as the authors are claiming that IL-35 treatment increases the proportion of anti-inflammatory macrophages. The effect that the authors are showing may be an indirect effect of IL35 and there might not be a causal relationship between IL35 and M1/M2 proportions. This point could easily be addressed by performing an in vitro experiment.
Author Response
The ICS TNFalpha is not reliable and it needs to be performed on cells restimulated and treated with GolgiStop. The restimulation is usually carried over for 3-4 hrs and in this short time-frame, no changes in the macrophage phenotype will occur.
Authors: We agree with reviewer’s concern. As we mentioned in our previous rebuttal letter that macrophages are highly plastic in nature, thus, we didn’t stimulate the cells. Moreover, in the present study we are comparing the population of M1 and M2 in STZ mice with the control mice.
The evaluation of the direct effect of IL35 on macrophage polarization is not trivial, as the authors are claiming that IL-35 treatment increases the proportion of anti-inflammatory macrophages. The effect that the authors are showing may be an indirect effect of IL35 and there might not be a causal relationship between IL35 and M1/M2 proportions. This point could easily be addressed by performing an in vitro experiment
Authors: We agree that this a very good suggestion. In the manuscript we don’t claim that the effect seen in this study of IL-35 on macrophages is directly due to IL-35. This has been clarified in the discussion line numbers: 170-173 . Also, with the given time for revision or revised the manuscript is not sufficient for performing such experiments considering the delivery time of required reagents specifically recombinant IL-35 (it can be taken up to 3 months due to present circumstances). Moreover, we didn't recover a sufficient number of live macrophages to perform such experiments in one of our pilot study.
Round 4
Reviewer 2 Report
The authors have not improved the manuscript, and none of the issues raised in the previous rounds of revisions have been addressed